# Monitoring a Bolted Vibrating Structure Using Multiple Acoustic Emission Sensors: A Benchmark

**Emmanuel Ramasso** * , **Benoît Verdin and Gaël Chevallier**

FEMTO-ST, Department of Applied Mechanics, 24 Rue de l'Epitaphe, 25000 Besançon, France; benoitverdin@hotmail.com (B.V.); gael.chevallier@univ-fcomte.fr (G.C.)
* Correspondence: emmanuel.ramasso@femto-st.fr

**Abstract:** The dataset presented in this work, called ORION-AE, is made of raw AE data streams collected by three different AE sensors and a laser vibrometer during five campaigns of measurements by varying the tightening conditions of two bolted plates submitted to harmonic vibration tests. With seven different operating conditions, this dataset was designed to challenge supervised and unsupervised machine/deep learning as well as signal processing methods which are developed for material characterization or structural health monitoring (SHM). One motivation of this work was to create a common benchmark for comparing data-driven methods dedicated to AE data interpretation. The dataset is made of time series collected during an experiment designed to reproduce the loosening phenomenon observed in aeronautics, automotive, or civil engineering structures where parts are assembled together by means of bolted joints. Monitoring loosening in jointed structures during operation remains challenging because contact and friction in bolted joints induce a nonlinear stochastic behavior.

**Keywords:** acoustic emission; bolted joints; health monitoring; laser vibrometry; benchmarking

## 1. Summary

Acoustic emission (AE) is one of the most promising nondestructive techniques for its ability to detect changes in the integrity of materials at the microscale. When damage occurs, an elastic wave is released which propagates onto the surface, creating displacements ranging from picometers to nanometers. The displacements are detected by highly sensitive piezoelectric sensors permanently attached onto a material. Data are continuously collected at sampling rate around 5 MHz. The obtained AE data stream (time-series) is then processed by algorithms to detect AE signals related to damages.

AE is used in many laboratories for materials characterization [1] and in industrial applications for real-time monitoring of manufacturing process [2] or storage facilities [3]. Despite many publications on the topic, the precise identification of the source of AE signals remains a challenge. Indeed, AE source identification is an inverse problem which is difficult to solve due to the sensitivity of the sensors which provide many AE signals corrupted by noise, as well as due to the effects of wave propagation in damaged materials which creates wave scattering [4].

The lack of prior knowledge on AE sources compels AE users to interpret AE by means of unsupervised learning algorithms which include the following steps:

1. AE signals detection: Also called wave-picking (similar to in seismology [5]), this step aims at processing the AE data stream to keep only relevant signals, defined as being above a given threshold (minimum amplitude). A preprocessing step can be included using wavelet denoising in noisy cases [6,7].

2. AE signals clustering: The signals obtained in the previous step feed a clustering method which aims at creating groups (clusters) of AE signals [8].

3. Clustering validation: The validation of the results obtained in the previous step is performed in a subjective manner because of the lack of prior knowledge of AE sources. To circumvent this problem, AE users apply different clustering strategies in the previous step and then compute clustering validity indices [9,10] which provide a scalar value that allows end-users to compare different results. The results can also be visually analyzed, such as in interactive clustering [11], to evaluate whether the clusters can be related to AE sources with a physical meaning [1].

An alternative to this three-step strategy is based on anomaly detection [12,13]. A baseline is considered to train a one-class supervised learning algorithm which can then be used to generate a health indicator for online monitoring [14]. Anomaly detection was also exploited for clustering in [15].

In real applications or for complex materials, AE signals detection (step 1), clustering (step 2), or validation (step 3), can be difficult. In that case, specific algorithms are developed. However, there are no common AE datasets that can serve as references (benchmark) to compare the methods of the literature. This work presents a benchmark dataset that can be used for this purpose.

The experiment was designed to reproduce the loosening phenomenon observed in aeronautics, automotive, or civil engineering structures where parts are assembled together by means of bolted joints. During the service life of a bolted structure, the torque can evolve according to operational conditions such as loading, in particular when these structures are submitted to vibrations. Monitoring the loosening condition in bolted structures during operation remains challenging because contact and friction in bolted joints induce a nonlinear stochastic behavior [16–21]. A common solution consists of measuring the pretightening force using a strain gauge bonded on the top surface of a bolt head [22]. This is a local approach since we need to instrument every bolt. On the other hand, global techniques make use of only a few sensors for bolt condition monitoring. There are two categories of global techniques: active and passive approaches. Active approaches, such as lamb waves, were used in the past and consist of using a transmitter of high harmonics waveforms tuned to be sensitive to the changes in the contacts. The receiver is used to collect the waveforms after their propagation in the bolted structure. Machine learning can then be used to build an anomaly detector such as in [13,14]. Conversely, passive approaches do not use an external source, but the transmitter is actually the structure itself (if a damage occurs). Therefore, a continuous reading of the piezoelectric sensors is necessary in order to not miss any damage or event. The use of AE sensors for condition monitoring of bolts is not frequent. For example, in recent works [23,24], it was shown that high-sensitivity piezoelectric sensors used in this technique are able to detect AE signals despite low signal-to-noise ratio in bolted joints under vibration.

In the context of stationary and harmonic vibration tests, the bolts can be subject to self-loosening under vibrations, eventually leading to a critical failure of the assembly. Therefore, it is of paramount importance to develop sensing strategies and algorithms for early loosening estimation. The main characteristics of this dataset are the following:

- During the tests, the tightening level of one of the bolts was controlled to reach seven different values, from highly tightened to loosened. Therefore, the end-user knows that the tightening level has been set to a specific value for some periods of the data stream. This reference can be used either to train supervised (deep/transfer) learning methods (with possibly seven classes) and for unsupervised or semisupervised learning. This reference can thus be helpful to validate steps 2 and 3 in the aforementioned methodology.

- The dataset is made of raw acoustic emission data streams acquired by three different sensors and a laser vibrometer. Multisensor data fusion can thus be performed and evaluated using the reference. This can be helpful to improve algorithms in steps 2 and 3.

- The vibrometer data also allows users to evaluate the performance of wave-picking algorithms with low signal-to-noise ratio. This can be helpful to validate step 1 in the aforementioned methodology.

## 2. Data Description

The dataset was obtained on a test rig called ORION [24,25]. It is constituted of a jointed structure, dynamically loaded with a vibration shaker and monitored with acoustic emission, force, and velocity sensors.

For each series of measurements, four sensors were used simultaneously: a laser vibrometer and three different AE sensors (micro80, F50A, micro200HF), each with a given frequency band, all sampled at 5 MHz, yielding approximately between 1.4 and 1.9 GB of data. A total of five series of measurements were performed, denoted as measurementSeries_B, measurementSeries_C, measurementSeries_D, measurementSeries_E, and measurementSeries_F.

The organization of repositories and files is depicted in Figure 1 for series of measurements B. There is one folder for each series of measurements. The ORION-AE dataset is thus composed of five folders. For each series of measurements, there are seven subfolders corresponding to seven tightening levels: 60, 50, 40, 30, 20, 10, 05 cNm, except for measurementSeries_C for which 20 cNm is missing.

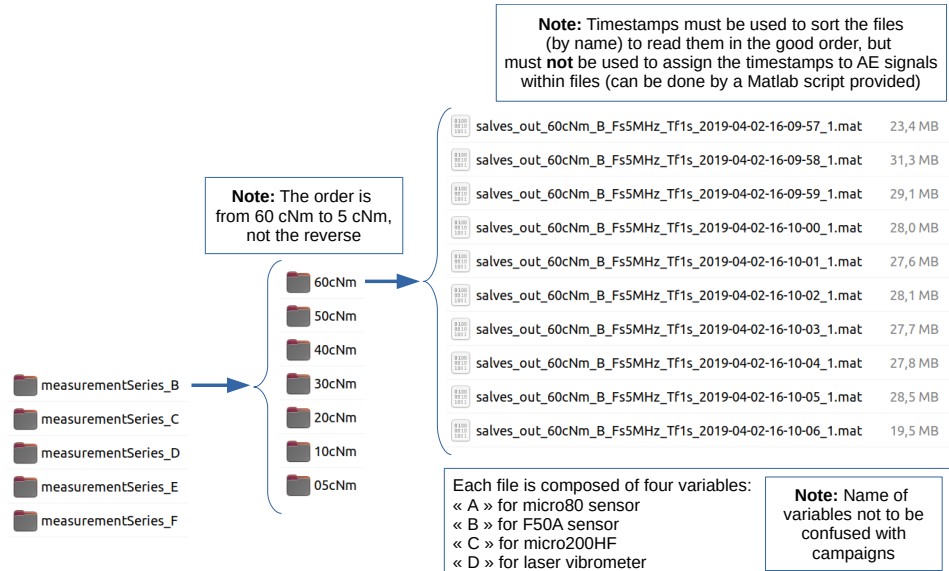

**Figure 1.** Tree structure of the repositories and files exemplified for series of measurements "B". The root directory on Dataverse also contains two additional repositories: One with images illustrating the content of the data in each level and campaign, and one with the sensors datasheet. Finally, a Matlab script is provided to read the files and correctly extract the timestamps.

It is important to read the subfolders in this precise order, otherwise the physical meaning of the data becomes incorrect. The upper bolts have been untightened with a precise sequence. For example, reading from 5 cNm to 60 cNm (reverse order) does not correspond to the "tightening" of bolts since the process is not reversible in terms of acoustic emission characteristics.

Each subfolder is made of .mat files generated using MATLAB 2016b. There is about one file per second. The files in a subfolder are named according to the timestamps (time of recording). Each file has the following format:

$$\text{salves\_out\_}\mathbf{X}\text{cNm\_}\mathbf{Y}\text{\_Fs5MHz\_Tf1s\_2019-04-02-HH-MM-SS\_1.mat}$$

where $\mathbf{X} \in \{5, 10, 20, 30, 40, 50, 60\}$ representing the tightening level, $\mathbf{Y} \in \{B, C, D, E, F\}$ for the series of measurements. The production date is "2019-04-02" and the timestamp began at HH-MM-SS. For example,

salves_out_05cNm_B_Fs5MHz_Tf1s_2019-04-02-16-22-09_1.mat

represents AE data for 5 cNm in series of measurements B taken at 16:22:09. Each .mat file is composed of vectors of data with the following names:

- Variable "A", corresponding to micro80 sensor data (sampling frequency: 5 MHz).
- Variable "B", corresponding to F50A sensor data (sampling frequency: 5 MHz).
- Variable "C", corresponding to micro200HF sensor data (sampling frequency: 5 MHz).
- Variable "D", corresponding to laser velocimeter data (sampling frequency: 5 MHz).

The number of samples is about $5 \times 4$ million per file (each file corresponds to about 1 s), with all sensors data sampled at 5 MHz. The labels corresponding to the sensors must not be confused with the labels corresponding to the test campaigns. Illustrations of the data are given in Figure 2 for sensor "F50A" (variable "B" in a file) in measurementSeries_B. About six periods are depicted (using the red curve representing the displacement of the upper beam).

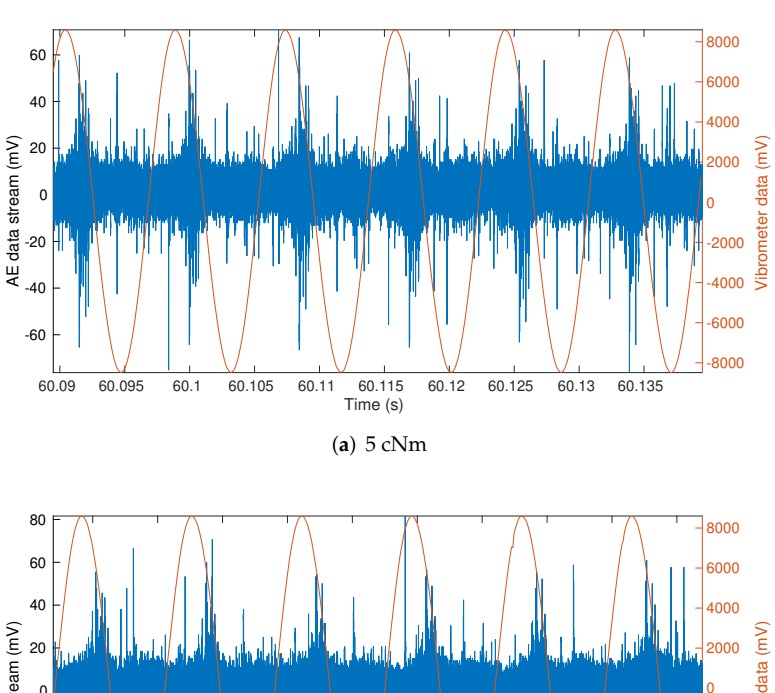

(**a**) 5 cNm

(**b**) 10 cNm

**Figure 2.** *Cont.*

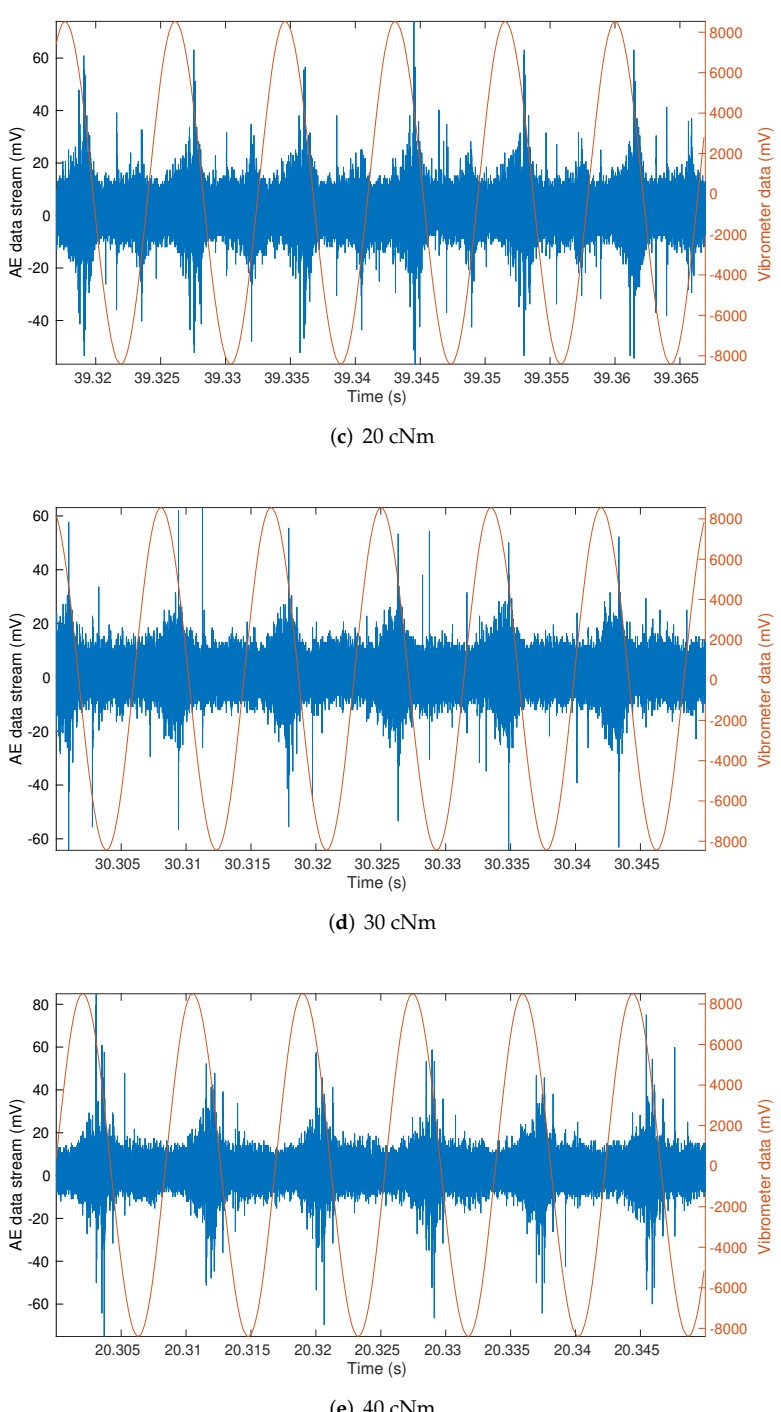

(**c**) 20 cNm

(**d**) 30 cNm

(**e**) 40 cNm

**Figure 2.** *Cont.*

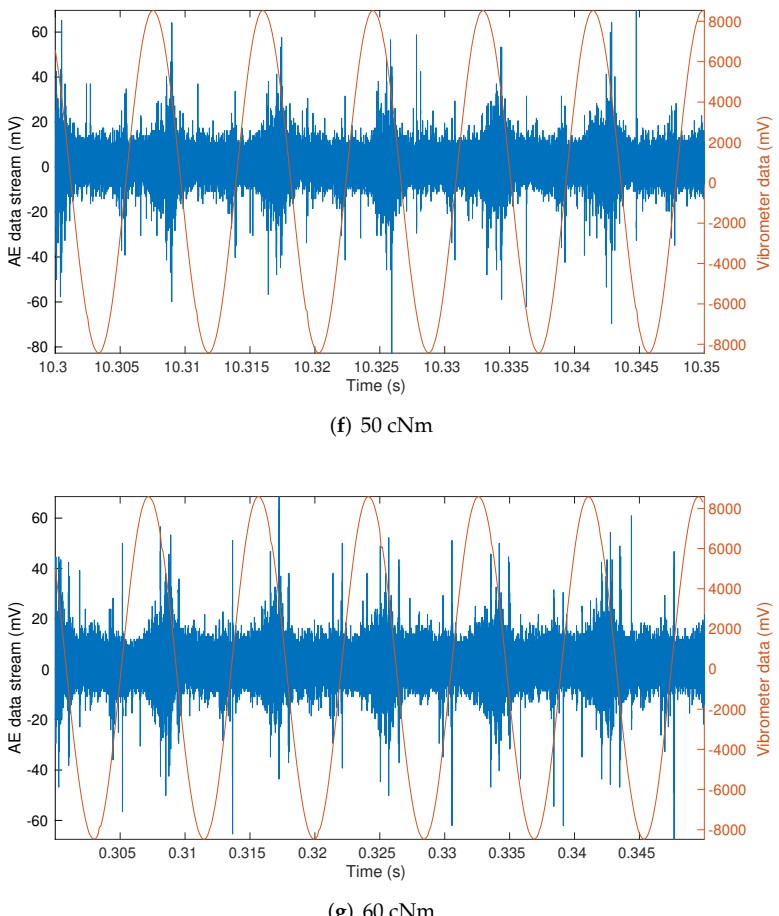

**(f)** 50 cNm

**(g)** 60 cNm

**Figure 2.** (**a**–**g**) Sample of the data in measurement series "B" with sensor "F50A" for each tightening level. Blue curve: AE signal; red curve: laser vibrometer data (harmonic vibration signal at 100 Hz with amplitude control).

Figure 3 depicts the acoustic emission and laser vibrometer data, superimposed with the labels (tightening levels) evolving from one to seven, for measurement *B*. This figure (red curve) shows that the control of the displacement of the beam by a feedback from the laser vibrometer, (as explained in the "Methods" section,) leads to a similar displacement during a whole test (about 70 s). In this figure, the green curve representing the acoustic emission shows that the amplitude cannot be used as a feature to discriminate the levels.

It is important to note that the real timestamps in filenames must not be taken as the timestamps of the AE signals in a data stream. The real timestamps must be calculated from the exact number of samples in each file, starting from $t = 0$ s for 60 cNm. There are about 10 s of continuous recording of data per level and for the four sensors. The exact duration of a period can be found according to the number of files in each subfolder, the number of points per file, and the sampling frequency. The precise values are given in Table 1. A Matlab code (ORION_AE_sample_read_files.m) is provided in the data directory to read the files and reproduce the duration of each period.

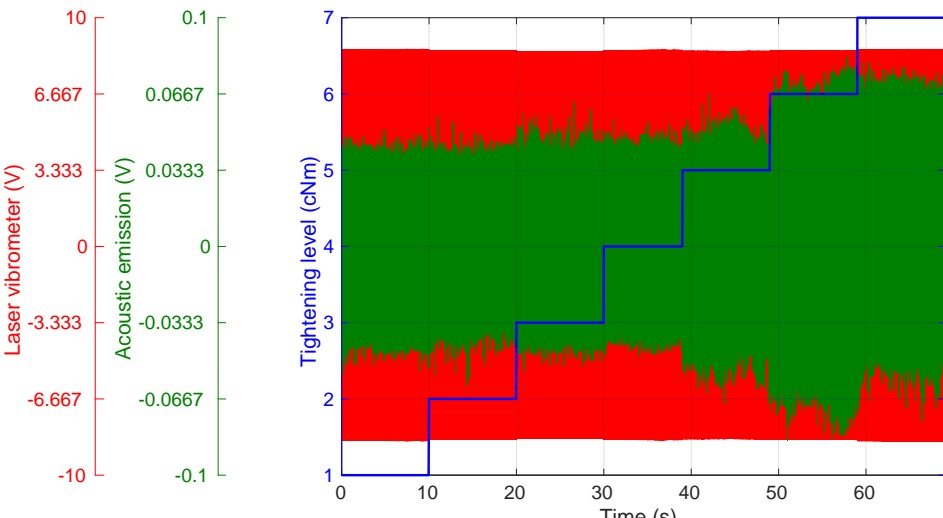

**Figure 3.** Tightening levels, acoustic emission, and laser vibrometer data superimposed for measurements "B" and sensor micro-200-HF (variable C). Due to the amount of data involved, a subsampling by a factor of 10 was applied for each file. The x-axis is here represented using the time of test (about 70 s), starting from 60 cNm from the left (around between $t \in [0, 10]$ s) to 5 cNm on the right (from $t > 60$ s).

**Table 1.** Duration in seconds of each period for a given tightening level. The exact duration of a period can be found according to the number of files in each subfolder, the number of points per file, and the sampling frequency. This table can be reproduced using a Matlab code (ORION_AE_sample_read_files.m) provided in the data directory.

| Meas. Series | 60 cNm | 50 cNm | 40 cNm | 30 cNm | 20 cNm | 10 cNm | 5 cNm |
|---|---|---|---|---|---|---|---|
| B | 10.0000 | 10.0000 | 10.0000 | 9.0170 | 10.0000 | 10.7725 | 9.2275 |
| C | 10.0000 | 10.0000 | 10.0000 | 10.0000 | N.A. | 10.0000 | 10.0000 |
| D | 10.0000 | 10.0000 | 10.8119 | 9.1881 | 10.0000 | 10.0000 | 10.0000 |
| E | 10.0000 | 10.0000 | 10.0000 | 10.0000 | 10.0000 | 10.0000 | 10.0000 |
| F | 10.0000 | 10.0000 | 10.0000 | 10.0000 | 10.0000 | 10.0000 | 10.0000 |

## 3. Methods

The ORION test rig (Figure 4) has been designed for the study of vibration damping and makes the vibration tests highly reproducible, as shown in [24,25]. It is a jointed structure made of two plates manufactured with 2024 aluminum alloy, linked together by three M4 bolts (Figure 4). Contact patches are machined overlays which have been added at each bolt connection to retain the contact between both beams in a small area, minimizing uncertainties on the structure's response and enhancing the repeatability between measurements. The contact patches have an area of $12 \times 12$ mm$^2$ and are 2 mm thick. The central bolt is dedicated to "static" functions, i.e., to ensure structural integrity and provide resistance to dynamic loads without substantial stiffness changes. The external bolts, in turn, perform "damping" functions, i.e., to increase energy dissipation due to frictional contact. It is worth noticing that these two functions have to be monitored in order to ensure the integrity (central bolt) and the damping (lateral bolts).

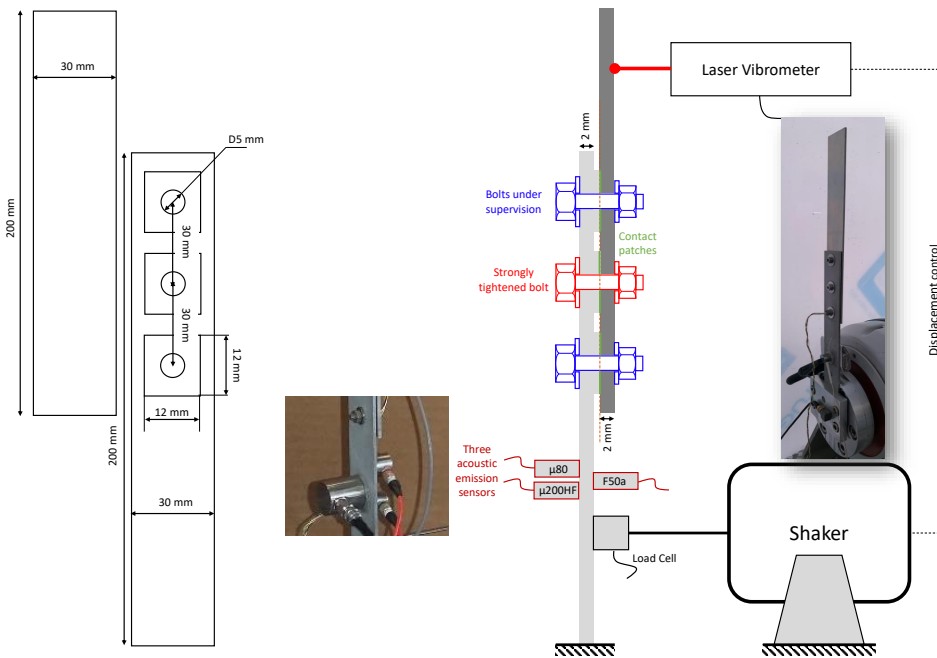

**Figure 4.** Setup description: part dimensions, sensors positions, bolts positions.

The harmonic excitation load was applied using a Tira TV51120 electromagnetic shaker which can deliver a 200*N* force. The structure was submitted to a 100 Hz harmonic excitation force for about 10 s. The force was measured using a PCB 208C02 piezoelectric force sensor.

The vibration magnitude was measured using a Polytec PSV500Xtra laser vibrometer. The obtained laser data allowed us to make sure that the amplitude of the displacement of the top of the upper plate remains constant for all tightening levels.

Seven tightening levels, corresponding to seven operating conditions of the structure under vibration, were applied on the upper bolt. The tightening level was first set to 60 cNm with a torque screwdriver. After about a 10 s vibration test, the shaker was stopped and this vibration test was repeated after a torque modification at 50 cNm. Then, torque modifications at 40, 30, 20, 10, and 5 cNm were applied. The change in the tightening level was made using a CDI torque screwdriver. The torque was checked three times at each change.

The acoustic emission sensors used were a "micro-200-HF", a "micro-80", and an "F50A", made by Euro-Physical Acoustics. They were connected to a preamplifer set to 60 dB (model 2/4/6 preamplifier made by Europhysical acoustics). Their detailed characteristics, such as dimensions and frequency bands, are provided in the data sheets enclosed in the dataset, with a summary of the main ones given in Table 2. The sensors were attached onto the lower plate (5 cm above the end of the plate) using silicone grease. All data were sampled at 5 MHz using a Picoscope 4824 (20 MHz bandwidth, low noise, 12-bit resolution, 256 MS buffer memory, USB 3) connected to a linux PC with MATLAB® 2016b and the Data Acquisition® toolbox. The data sheet of the Picoscope is provided in the dataset.

**Table 2.** Frequency and sensitivity of the acoustic emission sensors used. The data sheets are provided in the dataset.

|  | Units | F50A | micro80 | micro200HF |
|---|---|---|---|---|
| Peak sensitivity, Ref V/(m/s) | dB | 65 | 57 | 62 |
| Operating frequency range (kHz) | kHz | 200–800 kHz | 200–900 | 500–4500 |
| Resonant frequency, ref V/(m/s) | kHz | 100 kHz | 250 | 2500 |

## 4. User Notes

### 4.1. About the Ground Truth

In the ORION-AE dataset, we know that, during each period of 10 s, the tightening level has been set to a specific value. The tightening level can be considered as constant during each period, and therefore can be used as a reference. However, during each cycle of the vibration test for a given tightening level, different AE sources can generate AE signals and those sources may be activated or not, depending on the tribological conditions which are not known. The tightening level may thus slightly change during the period of 10 seconds. The tribological conditions may also be different from one campaign to another. Despite this variability, preliminary results shown in [26] demonstrated that the content of AE signals in those series of measurements is sufficient to discriminate between the periods, which seem to corroborate that, first, the AE sources have indeed a signature depending on the tightening level, and second, that the tightening level and the amplitude of the vibrations (controlled by a closed loop using the laser vibrometer) remain constant during the period of 10 s.

### 4.2. ORION-AE for AE Signal Detection

In addition to machine-learning tasks, this dataset can be used for signal processing and, in particular, wave-picking (step 1, presented in introduction). As for seismic wave detection [27], this step aims at finding relevant signals in a data stream. Wave-picking is one the most important tasks in acoustic emission analysis because it paves the way for feature extraction, classification, localization of acoustic emission sources, and interpretation by machine learning.

In ORION-AE, the AE sources are related to tribological phenomenon characterized by low amplitude and with low signal-to-noise ratio. Denoising methods are thus generally used as a preprocessing step. Figure 5 depicts the result of wavelet denoising with 14 levels of decomposition and using a Daubechies "db45" wavelet (the same settings as [7]). This was carried out for two sensors in the same chunk of data as shown in Figure 5a,b. It can be observed that the responses of the sensors differ because they have different characteristics (in particular the sensitivity and frequency band). We can also observe the presence of AE signals that occur at each cycle with some regularity.

We took the starting time, called onset, of each signal detected after wavelet denoising (from Figure 5 using [7]). Onsets were then reported onto the vibrometer data in Figure 6. Each blue circle corresponds to one AE signal detected. The vertical position represents the amplitude of the displacement (measured by the vibrometer) when the signal was detected. We can observe that the amplitude is very similar between cycles (there is about 1 s of test represented in this figure), which means that the activation of the AE sources is highly reproducible between cycles. Similar results have been observed on other levels and considering measurements B to F. However, we also observed that several AE sources can occur in some levels, and that the average amplitude of displacement can change. It means that the instant of activation of the AE sources (related to the signals) seems to depend on the tightening levels. These observations can be dependent on the wave-picking algorithm used (here [7]), and should be confirmed in the future.

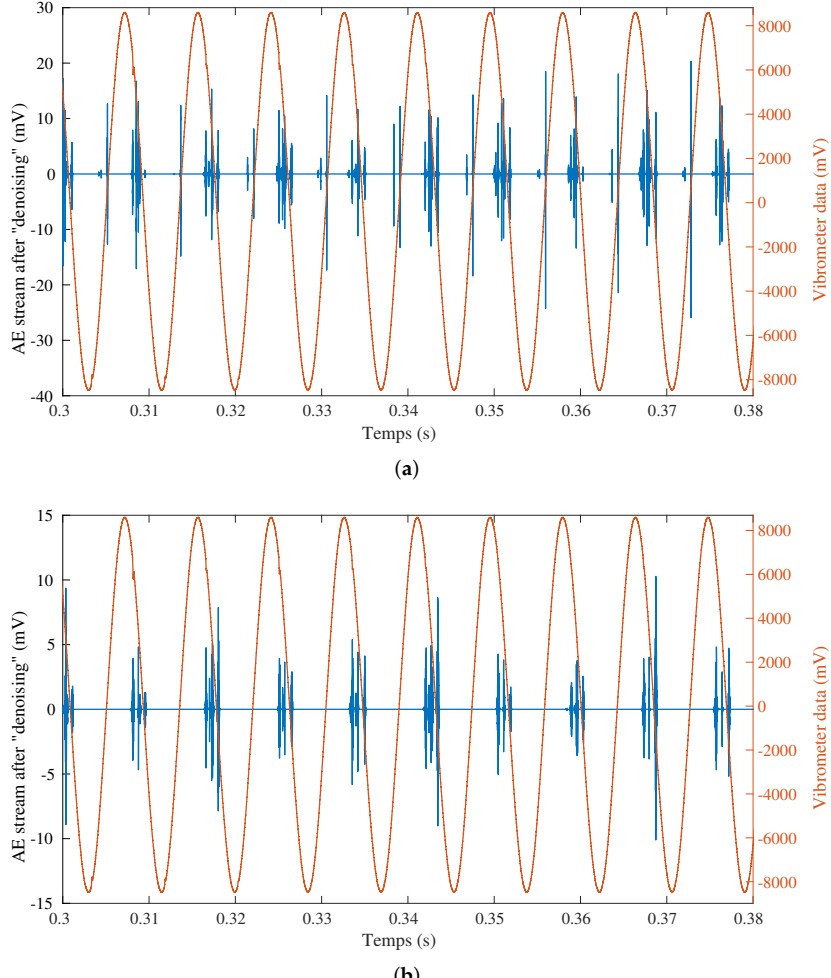

**Figure 5.** Presence of signals in each cycle after wavelet denoising using two sensors (on the same period of time). For a given tightening level, the displacement measured by the vibrometer is quite close for each signal found at different cycles showing the reproducibility of the test rig. (**a**) After wavelet denoising: Measurement B, first file in 60 cNm, sensor "F50A" (sensor 2). (**b**) After wavelet denoising: Measurement B, first file in 60 cNm, sensor "micro-200-HF" (sensor 3).

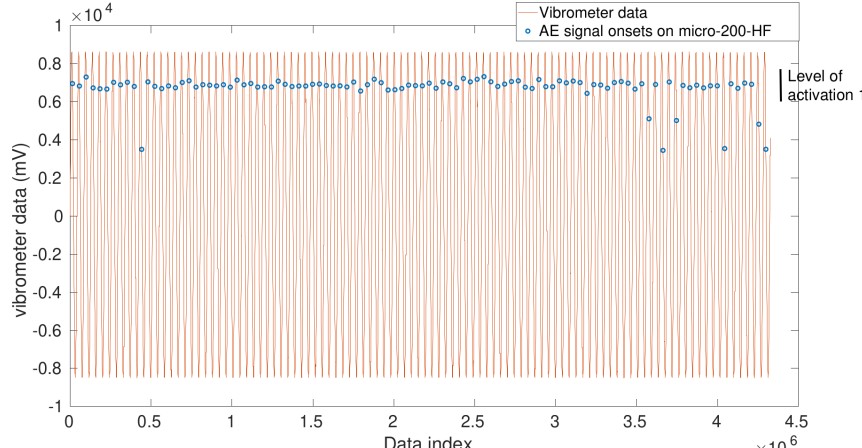

**Figure 6.** Acoustic emission onsets found by [7] and positioned onto vibrometer data for measurement B, first file in 60 cNm, sensor "micro-200-HF". The vertical position of the blue circle corresponds to the amount of displacement required to activate the AE sources. A similar amount is found in each cycle, which illustrates the reproducibility of the test rig.

### 4.3. Notebooks and Livescripts

Illustrations will be gradually provided at the following URL https://github.com/emmanuelramasso/ORION_AE_acoustic_emission_multisensor_datasets_bolts_loosening using livescripts in MATLAB or Python notebooks.

### 4.4. Possible Challenges

The ORION-AE dataset can be used for different tasks.

#### 4.4.1. Train Supervised Learning Methods

The classification task can consider up to seven classes in order to discriminate between the levels using various machine and deep learning methods such as support vector machines or neural networks [28,29]. By considering only the tightened levels, such as 60 cNm and 50 cNm, it can also be used to evaluate anomaly detectors based on autoencoders, self-organizing maps, or one-class classifiers [13,14]. In both cases, it is possible to use convolutional neural networks (CNN), taking as inputs the RGB images computed by the continuous wavelet transform (CWT) of the raw signals in each cycle. Cycles can be precisely obtained by using a zero-crossing detector on the vibrometer data. Illustration of such CWT images are depicted in Figure 7 using two measurement series (B, D).

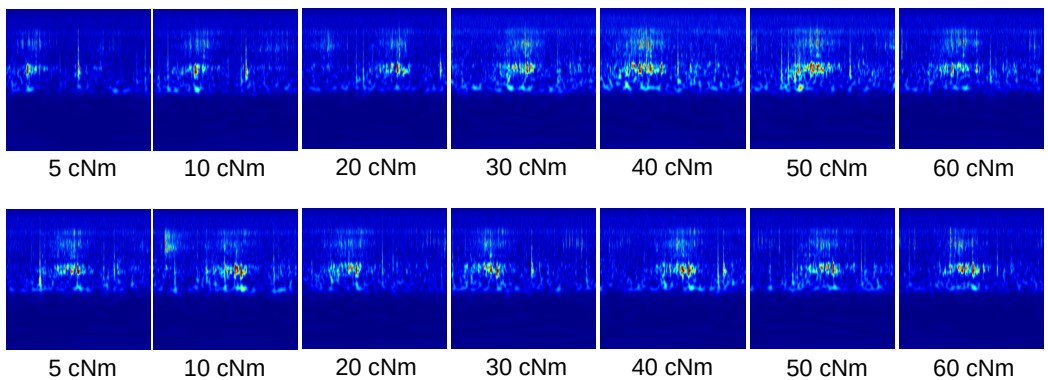

**Figure 7.** Continuous wavelet transform (CWT) images extracted for some cycles in sensor "F50A" (vector B) in measurement B (top figures) and in measurement D (bottom figures). Extraction of cycles can be automatically made by detecting zero-crossing in the vibrometer data (vector D).

#### 4.4.2. Data Normalization

Normalization aims at compensating time-varying operational variations which can occur in a system. According to our experience on this dataset, it seems that, for a given tightening level, the signature of the signals across campaigns seems to change. This can be due to the tribological conditions in the contacts that change between levels and campaigns. The classification algorithms (using deep learning and autoencoders) that we tested seem to work well when we mix levels and campaigns. However, the classification performance drastically drops when we distinguish the campaigns (for example training on B, C, D, and E, and testing on F is supposedly unknown, as in a SHM task). A data normalization is thus needed (starting from data in the first levels, for example 60 cNm or {60, 50} cNm). This is an important topic in SHM as presented in [14] (Chapter 12).

#### 4.4.3. Validate Unsupervised Learning (Clustering)

Clustering can be validated using the tightening levels as a reference to be compared with, using, for example, the adjusted Rand index [30]. Initial results on this dataset can be found in [26] using various algorithms.

#### 4.4.4. Challenge Wave-Picking Algorithms

Wave-picking algorithms can be challenged by trying to find either one set of signals per cycle or a set of several signals per cycle which should occur at similar displacement level along cycles for a given tightening level. Some results have been presented in the previous paragraphs using the method [7].

#### 4.5. Potential Sources of Error or Variability

According to our experience on this dataset, the low signal-to-noise ratio represents a real challenge to solve the aforementioned tasks. Some preliminary results have been shown in [26] with a good performance of various clustering methods, by considering campaigns independently. It shows that the information held by the data is relevant for estimating the tightening levels.

The variability across campaigns can represent a difficulty for simple classification methods, as discussed in the previous section. To tackle this issue, a proper data normalization is needed.

In terms of error in the dataset, we are aware that one level is missing: 20 cNm in campaign C. This means that this campaign is made of six classes.

This dataset was already used by master's students in 2021 and 2022 for machine learning illustration and an SHM course, at both the University of Franche-Comté and National Engineering School of Mechanics and Microtechnics of Besançon, who reported only one issue concerning the presence of a directory "5cNm" (made of one file) in addition to "05cNm" (made of nine files). This folder appears after the downloading. We recommend to delete the former after having transferred the file in the latter. This "phantom" directory is a small issue reported to the support of Harvard Dataverse and will be solved.

Finally, we acknowledge that this dataset is not representative of all possible datasets that can be collected using the AE technique. The purpose of this dataset is simply to provide a common reference to researchers in the AE community interested in developing new machine/deep learning or pattern recognition methods or new algorithms dedicated for AE data interpretation. With the expansion of deep learning, we believe that this dataset can be useful for benchmarking, which can lend a hand in discovering new and better ways of processing and interpreting acoustic emission data.

**Author Contributions:** Conceptualization, G.C. and E.R.; methodology, G.C. and E.R.; software, all authors; validation, all authors; investigation, all authors; resources, B.V.; data curation, B.V. and E.R.; writing—original draft preparation, E.R.; writing—review and editing, E.R. and G.C.; visualization, E.R.; supervision, E.R. and G.C.; project administration, E.R.; funding acquisition, E.R. All authors have read and agreed to the published version of the manuscript.

**Funding:** This work was partly carried out in the framework of: EIPHI Graduate school (contract ANR-17-EURE-0002), project RESEM-COALESCENCE funded by "Institut de Recherche Technologique" Matériaux Métallurgie Procédés (IRT M2P) and "Agence Nationale de la Recherche" (ANR); project CLIMA funded by the "Fond Unique Interministériel" (FUI 19).

**Institutional Review Board Statement:** Not applicable.

**Informed Consent Statement:** Not applicable.

**Data Availability Statement:** Directory name: Harvard Dataverse. Data identification number: 10.7910/DVN/FBRDU0. Direct URL to data: https://doi.org/10.7910/DVN/FBRDU0, comprising a Matlab code to read the files. Codes and main results will be updated on https://github.com/emmanuelramasso/ORION_AE_acoustic_emission_multisensor_datasets_bolts_loosening, accessed date: 12 May 2021.

**Acknowledgments:** The authors are also thankful to MIFHySTO and AMETISTE platforms.

**Conflicts of Interest:** The authors declare no conflict of interest.

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
