# Peer review of "Monitoring a Bolted Vibrating Structure Using Multiple Acoustic Emission Sensors: A Benchmark"

_data, 2019_

Round 1

Reviewer 1 Report

This research provides a benchmark study for comparing data-driven methods for bolt untightening monitoring using the measured acoustic emission (AE) data. The manuscript is well written with all the results clearly presented and in-depth discussed. Therefore, it is recommended for publication. Please kindly see below for some minor comments.

  1. Please consider revising the title to reflect the major task/contribution of this study. It is suggested not using the “dataset” as the subject of the title.
  2.     " ‡ These authors contributed equally to this work." Please indicate the authors with equal contribution, if any, in the author list.
  3. In figure 1, it is recommended to remove the red triangles which may be distractive. A red exclamation sign may be sufficient to attract attention to the notes.
  4. In Line 120, please make sure the MATLAB code is provided in the repository and it can be easily found by readers.

Author Response

Our responses are in the enclosed pdf file.

Reviewer 2 Report

Overall it is a useful dataset which is worth to be shared with the community. Here are my comments:

  • a bit more background information on the importance of monitoring jointed bolts is helpful.   
  • in the Summary section, the authors mentioned 3 steps towards using unsupervised learning (clustering) for analyzing AE data. I think another alternative approach should also be mentioned and it is through using unsupervised model such as deep autoencoders (deep learning) and anomaly/damage/defect detection through designing a threshold. Such an approach is now used in the literature for tool condition monitoring in machining processes. Please refer to the following paper in your manuscript where deep learning autoencoders for health monitoring is introduced and discussed (section 4.1 - Fig 10)

Nasir, V., Sassani, F. A review on deep learning in machining and tool monitoring: methods, opportunities, and challenges. Int J Adv Manuf Technol 115, 2683–2709 (2021). https://doi.org/10.1007/s00170-021-07325-7

  •     sampling frequency for all sensors should be mentioned in the manuscript. 
  • I am confused with the placement of three sensors on the set up. please clarify. 
  • the suggested citation and the threshold-based damage/anomaly detection method can also be included in section 4.3 of your manuscript where you are introducing some potential applications of your data.
  • in section 4.3 you wrote: "Illustration of CWT images that can be 199
    used as input of deep learning models are depicted in Figure 7. " This is not well-elaborated. Figure 7 is not explained well. you may clarify that CWT is a tool to generate image dataset and be linked to models such as CNN. 
  • explanation for Figure 6 is not very clearly presented. 
  • potential source of error or variability in your data can be mentioned (if any)
  • at this stage it is not very clear if machine learning  better model your data or deep models. you may introduce the potential methods for both unsupervised (autoencoders, SOM, clustering methods) and supervised (CNN, support vector, ANNs, etc) methods and make some discussion.  

Author Response

Please find the responses in the enclosed pdf file.

Round 2

Reviewer 2 Report

I appreciate the authors for addressing my comments.